# The Evolution of Landscape Patterns and Its Ecological Effects of Open-Pit Mining: A Case Study in the Heidaigou Mining Area, China

**DOI:** 10.3390/ijerph20054394

**Published:** 2023-03-01

**Authors:** Yuxia Zhao, Yang Wang, Zifan Zhang, Yi Zhou, Haoqing Huang, Ming Chang

**Affiliations:** 1Haikou Marine Geological Survey Center, China Geological Survey, Haikou 570100, China; 2Department of Architecture, University of Florence, 50121 Florence, Italy; 3Natural Resources Comprehensive Survey Command Center, China Geological Survey, Beijing 100055, China; 4Research Center of Applied Geology of China Geological Survey, Chengdu 610036, China; 5China Energy Information Technology Co., Ltd., Beijing 100011, China

**Keywords:** land use/cover, landscape pattern, ecological environment quality, remote sensing ecological environment index (RSEI)

## Abstract

This paper investigates the impact of land use/cover type changes in the Haideigou open-pit coal mine on the evolution of the landscape patterns and ecological and environmental quality in the mine area, based on medium- and high-resolution remote sensing images in 2006, 2011, 2016, and 2021 using ArcGIS 10.5, Fragstats 4.2, and the Google Earth Engine platform. The results show that: (1) From 2006 to 2021, the area of cropland and waste dumps in the Heidaigou mining area changed significantly, the land use shifted in a single direction, and the overall land use change was unbalanced. (2) Through the analysis of landscape indicators, it was shown that the diversity of the landscape patches in the study area increased, connectivity decreased, and the patches became more fragmented. (3) Based on the changes in the mean value of the RSEI over the past 15 years, the ecological environment quality of the mining area deteriorated first and then improved. The quality of the ecological environment in the mining area was significantly affected by human activities. This study provides an important basis for achieving the sustainability and stability of ecological environmental development in mining areas.

## 1. Introduction

Land use/cover change is considered to be a notable source of regional anthropogenic damage to the environment [1,2]. The activity of open-pit mining is a major source of dramatic damage to the surface landscape and the surrounding environment [3,4]. Open-pit mining is an activity that strips, transports, and accumulates the ground surface on a large scale [5,6]. The activity changes the landscape pattern through vegetation removal, soil denudation, and topographic reshaping [7,8,9]. It also affects the structure and function of the ecosystem. The activity of open-pit mining breaks the original balance of the ecosystem and eventually leads to a series of ecological and environmental problems, such as soil erosion, grassland degradation, desertification, ecological deterioration, environmental pollution, and geological disasters [10,11,12,13,14]. It seriously hinders global sustainable development and the construction of green mines in China [15,16].

With the development of remote sensing technology, land use changes in a study area can be detected in a timely and effective manner by using its features, such as multiple data, large range, fast update, and longtime data monitoring. The emergence of finer and higher-definition image technologies has led to the increasing accuracy of land use type classification and more precise detection of land use changes. In recent years, scholars from various countries have conducted much research on land use change, landscape pattern evolution, and ecological effects [17,18,19,20]. However, most of the research is about the effects of urban land use change on landscape and the ecological environment, and there are few studies on mining. The study by Garai et al. [21] shows that mining activities have a direct impact on the dynamic changes in land use types. Existing research results focus on the changes that occur in landscape patterns as a result of mining activities or land use change and an assessment of the ecological impacts [22]. For example, some studies on the spatial and temporal evolution characteristics of landscape patterns in large open-pit coal mining areas showed that changes in land use types in mining areas caused changes in landscape patterns, and landscape fragmentation and diversity increased accordingly [23,24,25,26]. Other studies have used remote sensing ecological indices to evaluate the ecological quality of mining areas, and the results show that land use changes also have a considerable impact on the ecological environment of mining areas [27,28,29,30]. Xiong et al. analyzed the spatial and temporal patterns and evolution characteristics of the ecological environmental quality in mining areas based on the Google Earth Engine [31]. This study highlights the advantages of using the GEE platform to monitor the ecological environment quality at the mining scale, not only for easy data acquisition and saving computation time but also for reliable results [32]. Previous studies have demonstrated that land use changes in mining areas are closely related to the landscape pattern of the mining areas and have a significant impact on the ecological environment [33].

Currently, most of the relevant studies focus on the following aspects: (1) Analyzing the spatial and temporal changes and driving forces of land use and landscape patterns in mining areas. (2) The response of ecological effects to land use change/landscape pattern evolution. (3) The interaction between land use systems and ecological service systems. However, relatively few studies have been conducted on the evolution of landscape patterns and ecological effects caused by land use changes in mining areas. In this paper, the evolution of landscape patterns and ecological effects caused by land use change in the Heidaigou open-pit coal mine are studied. The results of the study can provide an important scientific basis for land use planning in mining areas, establishing good ecological patterns and ecological environmental protection.

## 2. Materials and Methods

### 2.1. Study Area

The study area is the Heidegou open-pit coal mine in the Inner Mongolia Autonomous Region, which is rich in mineral resources (Figure 1a,b). The geographic coordinates are 111°10′00″–111°22′30″ east and 39°39′45″–39°44′15″ north (Figure 1c). The Heidaigou mine is 127 km south of Hohhot city and 120 km east of Ordos city, located in the northern part of the Ordos Loess Plateau, which is a typical loess plateau eroded hilly landscape. The topography of the mine area is high in the south and low in the north, with the general elevation ranging from 1100 to 1300 m. The highest point is located near Huangjialiang in the northwest of the mining area and has an elevation of 1308 m. The lowest point is located near the western boundary in the south of the mining area and has an elevation of 1093 m. The mine area belongs to the Yellow River system, which is a first-class tributary of the Yellow River. The Heidaigou open-pit coal mine is located in the semiarid continental monsoon climate zone in the middle temperate zone, with dry and windy springs, short hot summers, mild and pleasant autumns, and cold and snowless winters. The mine officially started production in 1999 and reached capacity the following year, with a design service life of 75 years and an estimated 1436 Mt of raw coal reserves to be mined (Figure 1d,e). In June 2006, after capacity expansion and renovation, the annual production of the Heidaigou open-pit coal mine reached 25 million tons of raw coal, making it the largest open-pit coal mine in China. In 2011, the annual production exceeded 31 million tons [34,35].

### 2.2. Data and Preprocessing

We used two data sources for this research. First, regarding the data source of land use, we used four phases of remote sensing image data, including Landsat TM data in July 2006, QuickBird data in August 2011, WorldView data in August 2016, and Jilin-1 satellite data in September 2021, with spatial resolutions of 30 m, 0.6 m, 0.5 m, and 0.8 m, respectively. Preprocessing, such as radiometric calibration, atmospheric correction, and image cropping, was implemented. To ensure the comparability of the images with different resolutions, the images of the four phases were resampled to a resolution of 10 m*10 m. We used the random forest (RF) classifier, which is based on machine learning algorithms and generally outperforms conventional classifiers, to classify and extract land use and land cover types in the study area using the ENVI software program; the algorithms’ parameter settings were demonstrated as default values.

Second, we used Landsat 5 TM and Landsat 8 OLI/TIRS images from the GEE platform. After selecting the best August image for the plant and climate conditions of the study area, we generated four spatial resolution 30 m RSEI maps for 2006, 2011, 2016, and 2021 by Landsat cloud mask, mean synthesis, and other algorithms.

### 2.3. Methods

#### 2.3.1. Land Use Transfer Analysis

The land use transfer matrix is a dynamic model of land use change. It can more intuitively describe the structural characteristics of land use change in study areas with changes in land use types, such as transfer direction, conversion source, and transfer rate. It is important for analyzing the spatial and temporal changes in land use. The mathematical expression of the land use transfer matrix is shown below:(1)$$\begin{array}{c} A=\left[\begin{array}{ccc} \begin{array}{cc} \begin{array}{c} A_{11}\\ A_{21} \end{array} & \begin{array}{c} A_{12}\\ A_{22} \end{array} \end{array} & \begin{array}{c} \cdots \\ \cdots \end{array} & \begin{array}{c} A_{1n}\\ A_{2n} \end{array}\\ \begin{array}{cc} \begin{array}{c} \vdots \\ A_{n_{1}} \end{array} & \begin{array}{c} \vdots \\ A_{n2} \end{array} \end{array} & \cdots & A_{nn} \end{array}\right] \end{array}$$
where A is the transfer matrix and Aij is the area of land use type I at time k transformed to land use type j at time k + 1 [36]. In this paper, the ArcGIS 10.5 platform was used to conduct a transfer matrix analysis of the land use in the study area from 2006 to 2021, focusing on comparing the spatial and temporal changes in land use in the open-pit coal mines over a long-term time series.

#### 2.3.2. Landscape Pattern Analysis

Landscape pattern is the type, number, and spatial distribution and configuration of landscape component units, which is the comprehensive spatial expression of landscape heterogeneity and reflects the spatial structural characteristics of the landscape [37,38,39,40]. To better understand the spatial and temporal changes in the landscape, the landscape pattern indices studied in this paper were divided into two categories. The first was the class metrics, including patch density (PD), patch area ratio (PLAND), landscape shape index (LSI), maximum patch index (LPI), and patch cohesion index (COHESION). The second category of landscape metrics included the Shannon diversity index (SHDI), the contagion index (CONTAG), and the modified Simpson uniformity index (MSIEI) [41,42].

(1)The Patch density

The patch density is the ratio of the number of patches of a certain type in an area, which shows the density of a certain type of patch in the landscape and can reflect the fragmentation of the landscape or the degree of fragmentation of a certain type.
(2)$$\mathrm{PD}=\frac{n_{i}}{A}$$
where PD is the patch density, PD > 0, $n_{i}$ is the number of patches in the landscape of patch type (class) i, and A is the area of the study area ($m^{2}$).

(2)The Percent Plaque Area

The percentage of patch area, also called the patch area ratio in some cases, is the ratio of various types of land types to the total area, with the largest area being the dominant landscape.
(3)$$\mathrm{PLAND}=P_{i}=\frac{\sum_{j=1}^{n}a_{\mathrm{ij}}}{A}\times 100$$
where $P_{i}$ represents the proportion of the landscape occupied by patch type (class) i , 0 < $P_{i}$ ≤ 100, $a_{\mathrm{ij}}$ represents the area ($m^{2}$) of patch ij, and A represents the total landscape area ($m^{2}$).

(3)The Landscape Shape Index

The landscape shape index is the shape index of patches in the landscape pattern. It is calculated by calculating the complexity of the shape of a patch in the area. A larger value for this index indicates a more complex landscape shape.
(4)$$\mathrm{LSI}=\frac{0.25\sum_{k\;=1}^{m}e_{\mathrm{ik}}^{*}}{\sqrt{A}}$$
where $e_{\mathrm{ik}}^{*}$ is the total length (m) of the edge in the landscape between patch types (classes) i and k and includes the entire landscape boundary and some or all background edge segments involving class i. A is the total landscape area ($m^{2}$). LSI ∈[1,+∞]. LSI = 1 when the landscape consists of a single square patch of the corresponding type; LSI increases without limit as landscape shape becomes more irregular and/or as the length of edge within the landscape of the corresponding patch type increases.

(4)The Largest Patch Index

The largest patch index (LPI) can help determine the dominant patch type in the landscape, i.e., the dominant patch; the index can also indirectly reflect the direction of disturbances caused by human activity.
(5)$$\mathrm{LPI}=\frac{a_{\max}}{A}\times 100$$
where $a_{\max}$ is the area of the largest patch ($m^{2}$) in the landscape or a patch type and A is the total landscape area ($m^{2}$). 0 < LPI ≤ 100.

(5)The Patch Cohesion Index

The patch aggregation force index measures the physical connectivity of the corresponding patch type.
(6)$$\mathrm{COHESION}=\left[1-\frac{\sum_{j\;=1}^{n}p_{\mathrm{ij}}{}^{*}}{\sum_{j\;=1}^{n}p_{\mathrm{ij}}{}^{*}\sqrt{a_{\mathrm{ij}}{}^{*}}}\right]\times {\left[1-\frac{1}{\sqrt{Z}}\right]}^{-1}$$
where p_ij_* is the perimeter of patch ij in terms of the number of cell surfaces, a_ij_* is the area of patch ij in terms of the number of cells, and Z is the total number of cells in the landscape. 0 < COHESION < 100.

(6)The Contagion Index

The contagion index describes the degree of aggregation or the tendency of extension of different patch types in the landscape. Since this index contains spatial information, it is one of the most important indices for describing landscape patterns.
(7)$$\mathrm{CONTAG}=\left[1+\frac{\sum_{i\;=1}^{m}\sum_{k\;=1}^{m}\left[P_{i}\times \frac{g_{i\kappa }}{\sum_{k\;=1}^{m}g_{\mathrm{ik}}}\right]\times \left[1n\left(P_{i}\times \frac{g_{i\kappa }}{\sum_{k\;=1}^{m}g_{\mathrm{ik}}}\right)\right]}{21n\left(m\right)}\right]$$
where P_i_ is the proportion of the landscape occupied by patch type (class) i, g_ik_ is the number of adjacencies (joins) between pixels of patch types (classes) i and k based on the double-count method, and m is the number of patch types (classes) present in the landscape, including the landscape border if present. 0 < CONTAG ≤ 100. CONTAG approaches 0 when the patch types are maximally disaggregated (i.e., every cell is a different patch type) and interspersed (equal proportions of all pairwise adjacencies). CONTAG = 100 when all patch types are maximally aggregated.

(7)The Shannon’s Diversity Index

The Shannon diversity index (SHDI) reflects the heterogeneity of the landscape and is particularly sensitive to the uniformity of the distribution of patch types in the landscape.
(8)$$\mathrm{SHDI}=-\overset{m}{\underset{i\;=1}{\sum }}\left(P\times {\mathrm{lnP}}_{i}\right)$$
where $P_{i}$ is the proportion of the landscape occupied by patch type (class) i. SHDI ∈ [0, +∞].

(8)The Modified Simpson’s Evenness Index

The modified Simpson evenness index reflects the evenness of the distribution of different types of patches in the landscape.
(9)$$\mathrm{MSIEI}=\frac{\mathrm{MSIDI}}{{\mathrm{MSIDI}}_{\max}}=\frac{-1n\sum_{i\;=1}^{m}P_{i}{}^{2}}{\ln m}$$
where Pi is the proportion of the landscape occupied by patch type (class) i and m is the number of patch types (classes) present in the landscape, excluding the landscape border if present. MSIEI ∈ [0, +∞].

#### 2.3.3. The Remote Sensing Ecological Index (RSEI)

The normalized difference greenness index (NDVI), humidity (WET), dryness (normalized difference building and soil index, NDSI), and heat (LST) were extracted by Xu [43,44,45,46,47,48,49]. Then, the remote sensing ecological index (RSEI), which reflects the state of the regional ecological environment, was derived by combining the wavebands and principal component analysis. The calculation process involved normalizing the four factors so that the results were mapped to the interval [0, 1]. Finally, the normalized indices were subjected to principal component analysis, and the original ecological index ${\mathrm{RSEI}}_{0}$ was constructed by the results of the principal component transformation. Its formula is as follows [50,51]:(10)$${\mathrm{RSEI}}_{0}=\mathrm{PC}\left[f\left(\mathrm{NDVI},\mathrm{WET},\mathrm{NDSI},\mathrm{LST}\right)\right]$$

To facilitate the measurement and comparison of the indices, the RSEI was normalized to the value of ${\mathrm{RSEI}}_{0}$ between 0 and 1. Finally, the remote sensing ecological index RSEI was obtained; the larger the value, the better the ecological environment quality and vice versa. To further quantify and visualize the RSEI, the RSEI index of each year was divided into 5 grades: excellent, good, moderate, fair, and poor, at an interval of 0.2. The detailed calculation process for each factor and RSEI is shown in Figure 2. 

## 3. Results

### 3.1. Land Use/Land Cover Change Characteristics

The total land use area of the Heidaigou open-pit coal mine is 6143.35 hm^2^. Referring to the national standard and combining the land use characteristics of the Heidaigou open-pit coal mine production, the land use/cover type of the coal mine was classified into nine categories, including forest, grassland, arable land, village, road land, open-pit mining site, waste dumps, other industrial and mining land, and pond water surface. The classification accuracy of the classified images was evaluated and the Kappa coefficients were all greater than 80%, meeting the accuracy requirements (Table 1). To visualize the land use changes in the Heidaigou open-pit coal mine, the land type shifts in the mine area from 2006 to 2021 were calculated using the ArcGIS 10.5 platform (Figure 3).

The transfer of land from 2006 to 2011 is shown in Table 2, in which the area of grassland, forest, pond, and water was transferred out more than transferred in. The area of forest transferred out amounted to 816.25 hm^2^, which was mainly transferred to 466.03 hm^2^ of waste dumps and 319 hm^2^ of cultivated land. The area of grassland decreased by 24.40 hm^2^, and the absolute transfer out was 60.53 hm^2^. The area transferred to waste dump was 24.71 hm^2^ or partly transferred to arable land and other industrial and mining land, with areas of 10.50 hm^2^ and 13.59 hm^2^, respectively. The waste dump area and other industrial and mining areas converted to grassland were 19.46 hm^2^ and 6.49 hm^2^, respectively. The entire area of the pond (0.31 hm^2^) was converted into a drainage field (Figure 4).

The area of grassland increased by 0.32% from 2011 to 2016, mainly from waste dumps and other mining sites, with areas of 17.60 hm^2^ and 15.75 hm^2^, respectively. The area of cultivated land increased by 7.13%, mainly from forest. The area of open quarry increased by 4.30%, mainly from 346.51 hm^2^ of waste dump and 351.76 hm^2^ of forest. The area of other industrial and mining land increased by 3.41%. In five years, the area of industrial and mining land (open-pit quarries, waste dumps, and other industrial and mining land) encroaching on grassland was 23.47 hm^2^. The area of industrial and mining land converted to grassland was 33.97 hm^2^. An area of 432.99 hm^2^ of forest was converted to industrial and mining land. The area of cropland converted to forest was 467.27 hm^2^ (Table 3, Figure 5).

Waste dumps were the most transferred in land use type from 2016 to 2021, which were mainly transferred from grassland, transportation land, mining sites and forest land. The most transferred out land use type is arable land, which decreases from 12.84% to 0.66%. And the transferred out area is 759.77 hm^2^, mainly to 701.10 hm^2^ of forest land and 63.05 hm^2^ of mining sites. In 5 years, 36.63 hm^2^ of grassland is transferred to industrial and mining land. The area of forest land converted to industrial and mining land was 678.74 hm^2^. But the total area of industrial and mining land converted to forest land was 66.39 hm^2^. (Table 4, Figure 6). 

In summary, all land use types in the study area were transferred in or out during the 15 years (Table 5). Among them, the situation of the forest, grassland, and agricultural land were as follows. The area of grassland fluctuated greatly; its area share decreased from 1.45% in 2006 to 0.98% in 2021, with the greatest decrease between 2006 and 2011. Its area decreased from 70.86 hm^2^ in 2006 to 60.24 hm^2^ in 2021, a decrease of 10.61 hm^2^, the area share decreased by 0.47%. Its main transfer out in the early stage was to the open pit, and the transfer in the later stages was mainly to the discharge field and woodland; its change location was concentrated in the southeast of the mining area (Figure 7). The area of woodland decreased from 3003.92 hm^2^ in 2006 to 2279.43 hm^2^ in 2021; the area transferred out was 724.49 hm^2^, accounting for 11.79% of the total area transferred out, which was mainly transferred out to open pit and waste dumps. Its change location was concentrated in the southeast of the mining area (Figure 7). The change in arable land area shows that the area decreased from 1110.45 hm^2^ in 2006 to 40.26 hm^2^ in 2021, and 17.42% of its area was transferred out to forestry and grass land types, mainly woodland. Its change location was concentrated in the southeast of the mining area (Figure 7). In absolute terms, the largest area of land use transferred out was 1070.19 hm^2^ of cropland, followed by 724.49 hm^2^ of forest.

Industrial and mining land, mainly open-pit quarries, waste dumps, and other industrial and mining land, increased yearly. Their areas increased from 50.30 hm^2^, 1513.12 hm^2^, and 318.10 hm^2^ to 806.93 hm^2^, 2504.90 hm^2^, and 392.97 hm^2,^ respectively, increases of 12.32%, 16.14%, and 1.22%, respectively. Among them, the largest area transferred was the open-pit quarry, with an area of 991.78 hm^2^. The next largest area was 756.63 hm^2^ of the wastedump. The main types of land transferred to industrial and mining land in the past 15 years were forestry land and agricultural land. In addition, almost all the water areas were transferred out to waste dumps; the area of villages decreased by 0.17% of the total area; the road land slightly fluctuated and increased slightly with the expansion of production activities. The conversion relationship between the various land use types is expressed in Figure 8.

From 2006 to 2021, the area of forest and grassland decreased significantly. This is mainly due to the increasing scale of coal mining and the occupation of forest and arable land by open-pit mines, waste dumps, and other industrial and mining land. Although the changes in land use/land cover type caused some ecological loss, in terms of area, the ecological loss was still within a reasonable range, and the greening and reclamation of the Heidaigou mining area were effective.

### 3.2. Landscape Pattern Change Analysis

#### 3.2.1. Landscape Pattern Change Characteristics at the Land Use/Cover Type Level

The landscape pattern indices at the land use/cover type level were analyzed for four periods from 2006 to 2021 in Heidaigou. The largest proportion of the landscape types in the Heidaigou mining area were forest, followed by waste dumps, open pits, and cultivated land. The areas of road land and villages were the smallest. From the landscape percentage data from 2006–2021, the area of forest landscape (mainly forest) and agricultural landscape (mainly cropland) decreased yearly, from 48.43% and 18.19% in 2006 to 31.41% and 0.64% in 2021, respectively. In contrast, the industrial type of landscape dominated by waste dumps and open-pit quarries gradually dominated, increasing by 6.45% and 12%, respectively (Figure 9a). In terms of the maximum patch index, the maximum patch index of grassland decreased and then increased during the 15 years, the maximum patch index of cultivated land did not change much, and forest showed a gradual decrease. The decrease in the maximum patch index indicates that the original concentrated natural landscape patch area decreased with increasing production activities (Figure 9b). The patch cohesiveness index showed that the spatial cohesiveness of all the landscape types was high. The cohesiveness of grassland and cropland fluctuated slightly, with the cohesiveness of the grassland decreasing and then increasing with an overall increasing trend, the cohesiveness of the cropland decreasing slightly and showing a decreasing yearly trend, and the cohesiveness index of the forest varying. The cohesiveness of the village area decreased, and the trend of connectivity gradually decreased. The cohesiveness of the road land was the smallest, but showed a trend of increasing yearly, indicating that the spatial connectivity of the transportation land was rising (Figure 9c). From the landscape shape index, the shape index of the cultivated land and forest decreased yearly, indicating that their shapes tended to be simpler and reflecting that the mine area underwent more land reclamation and vegetation restoration planning in recent years (Figure 9d). In terms of the patch density index, during the 15-year period the higher patch density in the mining area was for cropland and road land. However, the patch density of the arable land decreased each year. The patch density of the grassland, woodland, and village types showed similar trends, decreasing and then increasing, indicating that the landscape ecological processes were more active in the study area (Figure 9e).

#### 3.2.2. Landscape Pattern Change Characteristics at the Landscape Level

The landscape structure of the study area has changed dramatically in the last 15 years. The change in the Shannon diversity index showed an increasing trend from 2006 to 2016 and decreased after 2016. Overall, it increased from 1.32 in 2006 to 1.43 in 2021. The increase in the Shannon diversity index reflects the trend of increasing heterogeneity of landscape types in the study area. (8) The Modified Simpson’s Evenness Index of the study area fluctuated slightly, but the overall value was not high, indicating that the distribution of landscape patches in the study area was not highly uniform and varied greatly. The spreading index of the study area showed a trend of first decreasing and then increasing. The spreading index of the study area showed a decreasing trend from 2006 to 2011, reflecting the decrease in the connectivity of the landscape patches and the deepening of landscape fragmentation, and an increasing trend after 2011, reflecting the increase in the connectivity of landscape patches and the decrease in landscape fragmentation (Figure 9f).

### 3.3. Evaluation of the Ecological Environment Quality of the Heidaigou Open-Pit Mine

#### 3.3.1. The Remote-Sensing-Based Ecological Index (RSEI)

Based on the GEE platform, four indicators, the greenness, humidity, dryness, and heat of the mining area for four years, 2006, 2011, 2016, and 2021, were extracted and subjected to principal component analysis (PCA). The results are shown in Table 6, where the contributions of PC1 are 70.32%, 71.56%, 75.36%, and 74.68%, respectively. This indicates that the first principal component PC1 concentrated most of the characteristics of the four indicators. Therefore, PC1 can be used to construct the RSEI index. Among the four indicators of PC1, the values of greenness and humidity are positive and the values of dryness and heat are negative. This means that the greenness and humidity are positively correlated with ecological quality and the dryness and heat indices are negatively correlated with ecological quality. The absolute value of the dryness index was the largest, which means that the most influential of the four indices was dryness. Human construction activities were closely related to the dryness values, indicating that the ecological quality condition of the mining area was most closely related to the mining activities and industrial construction.

#### 3.3.2. Analysis of Spatial and Temporal Changes in the Ecological Environment of the Mining Areas

The ecological environment quality of the mining area over the 15 years is shown in Figure 10, where green, light green, yellow, orange, and red represent the ecological grades of ‘excellent’, ‘good’, ‘medium’, ‘fair’, and ‘poor’, respectively. Comparing the changes in the mean values of RSEI in the four periods of 2006, 2011, 2016, and 2021 (Table 7), it can be inferred that the overall change trend for the ecological environmental quality condition of the mining area first increased and then decreased. The mean values of the greenness and humidity indicators are consistent with the RSEI. This indicates that the ecological restoration effect of the mining area after 2011 was significant and that the ecological environment quality of the mining area improved. This result indicates that at the mine scale, a short period of human ecological restoration activities can rapidly improve the ecological environment quality situation.

The total area of the different grades of ecological environment quality in the study area for each period is shown in Table 8. The total area of the study area with ‘poor’ ecological environment quality over the last 15 years showed a rapid decrease in the previous period and reached its lowest value in 2016, with an area of 214 hm^2^. The percentage was reduced from 9.83% in 2006 to 3.48% in 2016. From 2016 to 2021, its area slightly increased to a proportion of 3.68%. The mining area’s share of ‘fair areas’ increased, with its sum reaching 1993.08 hm^2^ by 2021 and accounting for 32.44% of the mining area. The proportion of areas with ‘moderate’ ecological environment quality decreased, from 2552.47 hm^2^ in 2006 to 1964.94 in 2021, with the area proportion ranging from 41.55% to 31.98%. The mining areas with ‘good’ ecological and environmental quality reached a maximum in 2011, where it accounted for 36.13% of the total area. After that, there was a decreasing trend, with the lowest value in 2016 accounting for only 17.90% of the total; by 2021, the percentage had slightly increased to 19.61%. The percentage of the area with ‘excellent’ ecological quality increased from 307.95 hm^2^ in 2006 (5.01% of the total) to 728.71 hm^2^ in 2021 (11.86% of the total). In summary, there was a consistency in the changes for ‘excellent’ and ‘poor’ ecological environment quality areas in the study. That is, the ‘poor’ area decreases yearly, the excellent ‘area’ increases yearly, and the ‘fair’, ‘moderate’, and ‘good’ areas were very volatile.

The most significant change in the ecological quality occurred in the waste dump areas. The five waste dumps in the Heidaigou mining area were numbered A–E (Figure 11). The changes in the ecological quality of the waste dumps A, B, and C from 2006 to 2021 was similar. The ecological quality was the worst in 2006, then gradually improved, reaching a peak in 2011. The specific situation is that the area of ‘poor’ ecological environment quality of the waste dumps A, B, and C decreased sharply from 2006 to 2011, from 35.2% to 8.1%. The overall ecological and environmental quality of the three waste dumps was in a ‘good’ state. From 2011 to 2021, the ecological and environmental quality of the waste dumps A, B, and C did not change significantly and was relatively stable. The ecological quality of waste dump D first deteriorated and then improved from 2006 to 2021. The ecological quality was the worst in 2011. After that, it gradually improved. Waste dump D started to be used in 2006, so the area of ‘poor’ ecological quality expanded from 1.2% to 57.4% from 2006 to 2011. Thus, the ecological environment deteriorated. The ecological quality of waste dump D improved from 2011 to 2021, and the area of ‘poor’ ecological quality was only 5% by 2021. Waste dump E is an internal waste dump. Due to its special location next to the quarry, it is the one with the most worrying ecological environment quality among the five waste dumps. However, from 2006 to 2021 the ecological quality of waste dump E gradually improved and the area with ‘poor’ ecological quality decreased from 71.6% to 19.8% in 2021. The ecological quality of waste dump E is significantly better than at the beginning.

#### 3.3.3. Trends in the Changes of Ecological Quality

A ridge regression function was obtained using the GEE platform to fit the RSEI trends for every five years from 2006 to 2021 for the Heidaigou open-pit coal mine and the corresponding significance (Figure 12). The change in ecological quality is expressed as a numerical value: when the value ∈[−∞, −0.05], it indicates ‘significant deterioration’. When the value ∈[−0.05, −0.015], it indicates ‘deterioration’. When the value ∈[−0.015, 0.015], it indicates ‘no change’. When the value ∈[0.015, 0.05], it indicates ‘improvement’. When the value ∈[0.05, +∞], it indicates ‘significantly improved’.

The trend of ecological quality is reflected by the slope of the fitted ridge regression function. For example, a slope value greater than 0 indicates a positive direction for ecological change, whereas a value less than 0 indicates deterioration. The data were reclassified using ArcGIS to obtain five categories: ‘significantly optimized’, ‘getting better’, ‘unchanged’, ‘getting worse’, and ‘significantly worse’. From 2006 to 2011, the locations of waste dumps C and E in the northwestern part of the mine site deteriorated significantly. The ecological quality of waste dumps A, B, and C improved significantly. The overall ecological quality of the mine area changed as follows: the ‘significantly worse’ area accounted for 7.04%, ‘worse’ accounted for 28.03%, ‘unchanged’ accounted for 47.99%, ‘better’ accounted for 14.22%, and ‘significantly optimized’ accounted for 2.72%. The overall ecological environment quality of the mining area deteriorated (Figure 12a). From 2016 to 2011, the ecological quality of the mining area and the waste dumps became ‘significantly optimized’ and the percentage of the area that became ‘significantly worse’ and ‘worse’ was 31.29%. The sum of the area changes for ‘better’ and ‘significantly optimized’ was 33.87%. Due to the downward shift in the mining area, the ecological environment quality in the southeast of the mining area became ‘significantly worse’ (Figure 12b). From 2016 to 2021, the mining area and the waste dumps significantly increased their regional ecological quality. This indicates that the ecological quality of the area after artificial restoration became significantly better. It is noteworthy that the ecological quality of the waste dump significantly deteriorated in that period. This may be caused by inadequate maintenance work after reclamation (Figure 12c). The ecological quality of the mine area has improved in the last 15 years. The ecological quality of the artificially damaged areas, especially the waste dumps, were continuously being optimized.

## 4. Discussion

### 4.1. Analysis of the Reasons for the Landscape Evolution Patterns of the Open-Pit Coal Mine

The change in landscape patterns in the Heidaigou mining area from 2006 to 2021 were volatile, mainly showing that both forest-type landscapes (mainly grassland and forest) and agricultural landscapes (cultivated land) have decreased over the past 15 years. As the patch area of woodlands, the dominant landscape type, decreased, the variability between the dominant landscape and other types of landscapes shrunk, the landscape structure became more homogeneous, and the landscape fragmentation increased. The changes in the CONTAG (the spreading index), SHDI (the Shannon diversity index), and MSIEI (the modified Shannon evenness index) before 2011 reflect the increased diversity of landscape patches in the mining area, the deterioration of connectivity, and the increasing fragmentation of the landscape. The situation has improved since 2011. There are two main reasons for this change. First, the method of open-pit coal mining is mainly the stripping and excavation of the surface and the large volume of waste materials generated during the mining process are piled up on the surface, causing fragmentation of the landscape slab and easily aggravating the fragmentation of the landscape. Second, before 2011 the mine area focused more on production and ecological restoration and other efforts were slightly weaker. After 2011, the mining area started to focus on ecological work, land reclamation, vegetation restoration, and reforestation, especially in terms of reclamation while producing. These measures led to the opposite trend of landscape pattern change after 2011 and landscape fragmentation was improved.

### 4.2. The Ecological Evaluation of Mines through Analyzing Landscape Pattern Changes 

From the analysis of the landscape pattern changes and the RSEI index, it can be concluded that there was no correlation between the landscape pattern changes and the remote sensing ecological index. There are three main reasons for this: (1) The calculation results of the landscape index were more focused on reflecting the subtle changes in the land use types, which helped increase the proportion of benign land use/cover types in the mining area through the time series analysis of the landscape patterns. (2) The landscape pattern index was mainly calculated based on the land use/land cover types, and its land type classification determined the size of the landscape pattern index. The landscape types at the mine scale were more complex, the land use type classification was subject to greater human influence, and the ecological and environmental conditions of certain landscape types, such as the waste dumps, were polarized, so the changes in its landscape index could not be linked to changes in the ecological environment. (3) Third, the calculation of the landscape index was affected by the spatial granularity and magnitude of the study and the quality of the remote sensing data and different data processing methods, which resulted in errors in the data [52]. However, there was a correlation between the change in landscape percentages of industrial and mining land types at the individual level, such as open-pit mining, and the quality of the regional ecological environment.

However, the correlation between the landscape pattern changes and the ecosystem quality was not significant. However, a large number of studies at the landscape level in recent years have shown that it is closely related to ecosystem structure, function, and stability [52]. Some scholars believe that the intensification of anthropogenic activities promotes the development of landscape fragmentation and affects the energy flow and nutrient cycling in natural systems, leading to the reduction of ecosystem productivity and the loss of ecosystem functional stability. Anthropogenic activities change landscape patterns and directly affect the distribution and spread of species, resources, and anthropogenic disturbances in the landscape, e.g., urban expansion and mineral development [53]. This affects the biodiversity and productivity of ecosystems and plays an important controlling role in regional landscape functions and ecosystem processes. Other researchers use the landscape pattern index to construct landscape ecological risk indices and employ analytical methods such as spatial statistics and geostatistics to explore the spatial and temporal heterogeneity of the landscape ecological risk in mining areas at appropriate scales and to assess the driving forces of landscape ecological risk in mining areas. In addition, the analysis of landscape patterns was also considered to be closely related to primary productivity by Kang Sarula et al. [52]. Therefore, the analysis of landscape pattern changes in mining areas is of practical value for exploring ecosystem processes and studying ecological problems in mining areas.

### 4.3. Limitations of the RSEI Index in the Evaluation of the Coal Mine Ecological Environment

In recent years, the RSEI index has been widely used in the ecological environment evaluation work of mining areas. Li Rui et al. evaluated the ecological environment of the Shendong mining area from 1989 to 2016 based on Landsat images using the RSEI index [54]; Xia Nan analyzed the ecological environment of the Wucaiwan mining area from 2003 to 2015 based on Landsat and MODIS image ecological environment quality [55]; Han Li et al. analyzed the ecological environment quality of the Baorixile open-pit mine area from 1996 to 2019 based on Landsat images [56]. Compared with comprehensive ecological evaluation indices such as the ecological index (EI), its acquisition method is simpler and based on the GEE platform, which makes it easy to obtain a long-term series of ecological environment changes in mining areas. However, because the remote sensing ecological index is entirely based on remote sensing data, it leads to single data sources and over-relies on indicators such as greenness, humidity, dryness, and heat, which leads to its evaluation results being one-sided and ignores the complexity of ecological environment evaluation work. The small-scale RSEI index values are subject to large weather environment images, which also leads to the general reliability of its results from an objective aspect. Therefore, the ecological environment evaluation work of mining areas using the RSEI index needs to combine more ecological evaluation factors.

## 5. Conclusions

In this paper, the corresponding land use data were obtained by classifying four phases of images of the Haidaigou open-pit mine in 2006, 2011, 2016, and 2021 based on the random forest method. The characteristics of the dynamic changes in land use/land cover types were analyzed by using a transfer matrix. The landscape pattern indices of the Haidaigou mining area in each period were also analyzed. The evolution characteristics of the landscape pattern of the mining area were obtained and the ecological environment quality of the study area over the past 15 years was analyzed using the ecological environment index (RSEI). The main conclusions are as follows:Changes in land use/land cover types are important drivers of landscape pattern evolution and changes in ecosystem quality.The changes in the ecological index of the Heidaigou mining landscape has fluctuated significantly over the past 15 years, mainly due to the mining activities and land reclamation work in the mine and the mining method of the open-pit coal mine. The changes in the ecological environment quality of the mining area first decreased and then increased, which was mainly influenced by human activities; the measure of “reclamation while mining” had a significant effect on the protection of the ecological environment in the mining area.The correlation between the landscape pattern evolution and the changes in remote sensing ecological indices was weak, and research work on mining-scale landscape patterns should be more in the study of ecosystem structure, function, productivity, stability, and ecological risk.

## Figures and Tables

**Figure 1 ijerph-20-04394-f001:**
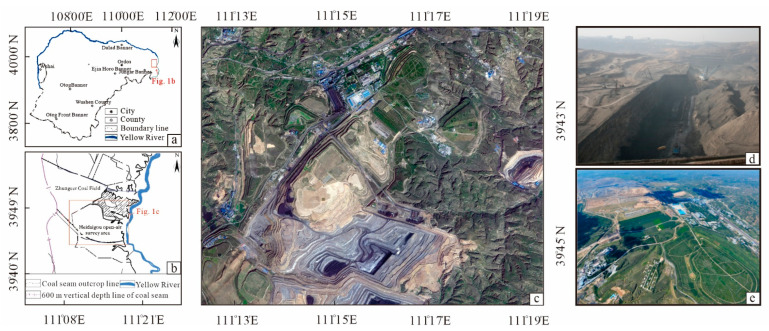
(**a**) The geographic location of the Jungar coalfield. (**b**) The geographical location of the Heidaigou mine area. (**c**) A remote sensing image map of the mine area in 2021. (**d**) A photograph of the mining site in Heidaigou. (**e**) The present situation of the slope greening of the waste dump in the Heidaigou mine.

**Figure 2 ijerph-20-04394-f002:**
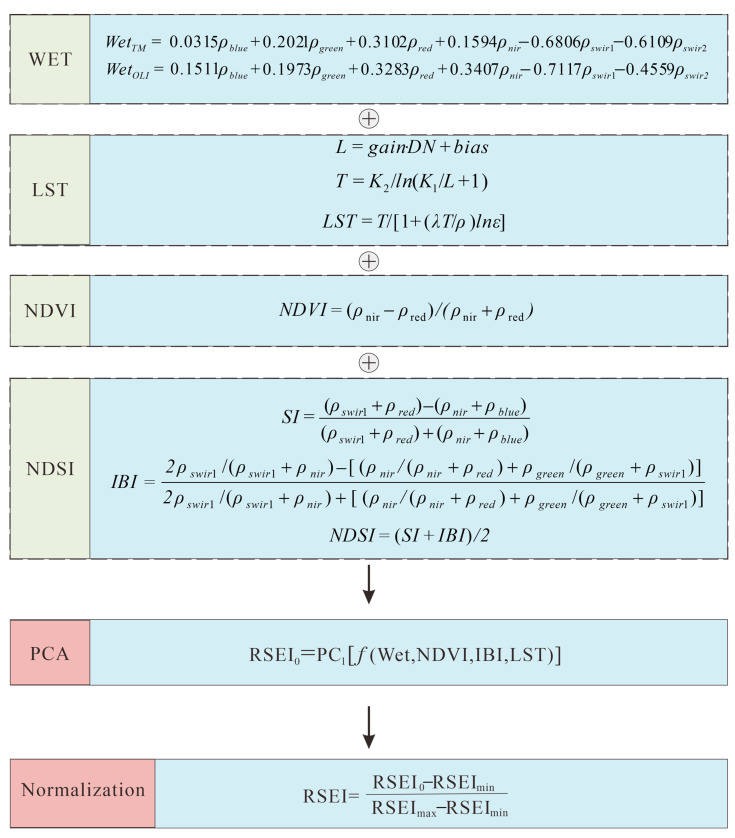
Flow chart of RSEI calculation.

**Figure 3 ijerph-20-04394-f003:**
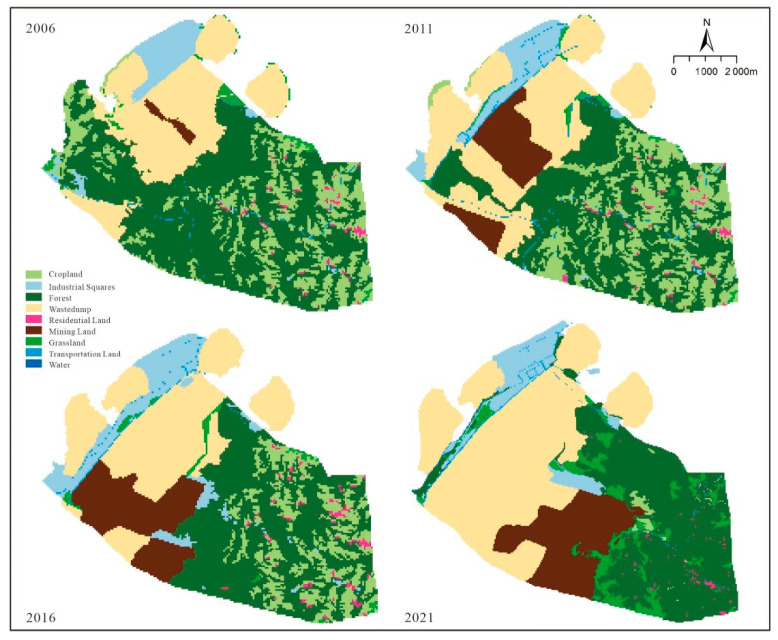
Land use/land cover type maps for 2006, 2011, 2016, and 2021.

**Figure 4 ijerph-20-04394-f004:**
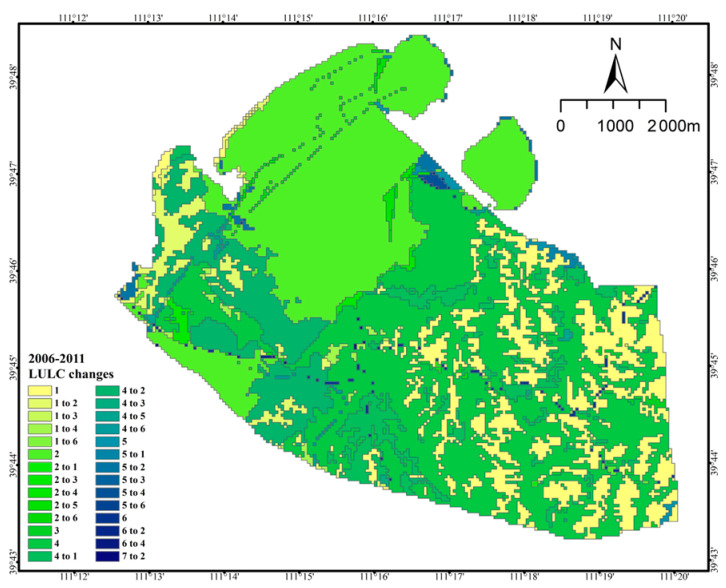
Land use/land cover type change from 2006 to 2011 (1. Cropland; 2. Industrial and mining land; 3. Forest; 4. Residential land; 5. Grassland; 6. Transportation land; 7. Waterbody).

**Figure 5 ijerph-20-04394-f005:**
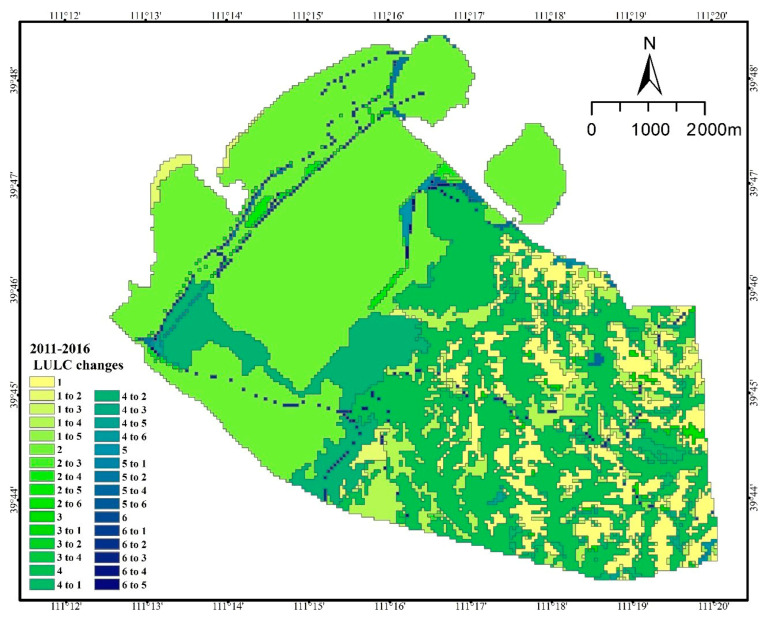
Land use/land cover type change from 2011 to 2016 (1. Cropland; 2. Industrial and mining land; 3. Forest; 4. Residential land; 5. Grassland; 6. Transportation land).

**Figure 6 ijerph-20-04394-f006:**
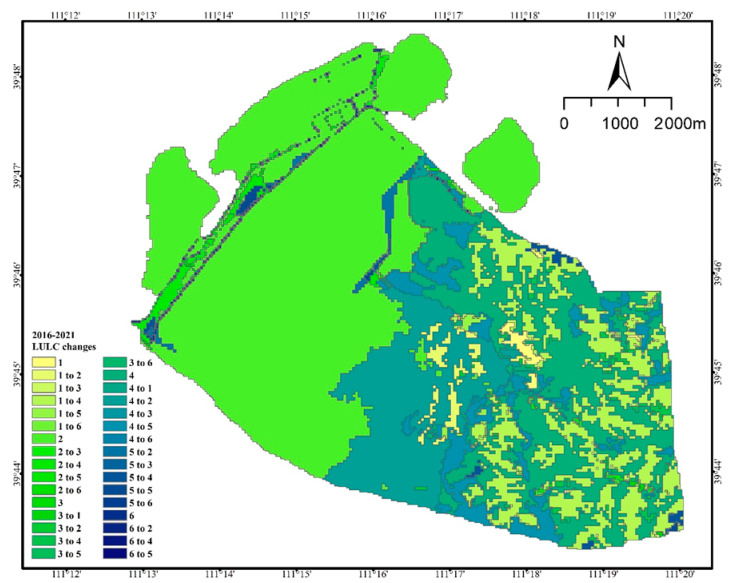
Land use/land cover type change from 2016 to 2021 (1. Cropland; 2. Industrial and mining land; 3. Forest; 4. Residential land; 5. Grassland; 6. Transportation land; 7. Waterbody).

**Figure 7 ijerph-20-04394-f007:**
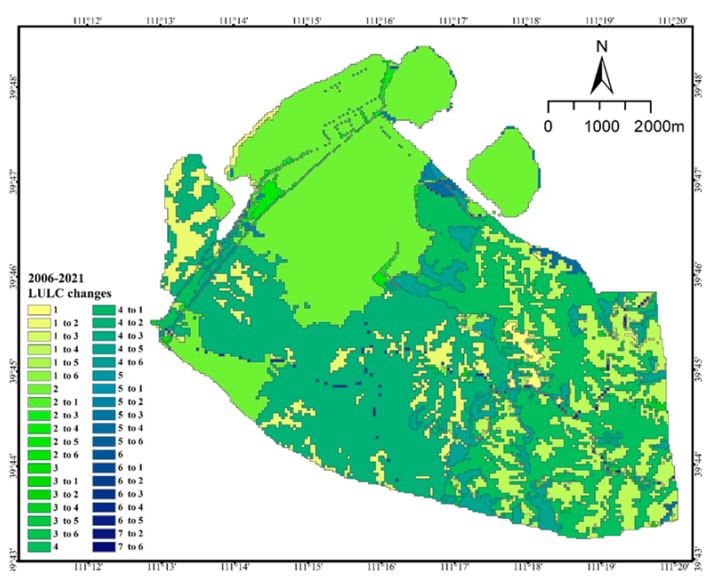
Land use/land cover type change for 2006 to 2021 (1. Cropland; 2. Industrial and mining land; 3. Forest; 4. Residential land; 5. Grassland; 6. Transportation land).

**Figure 8 ijerph-20-04394-f008:**
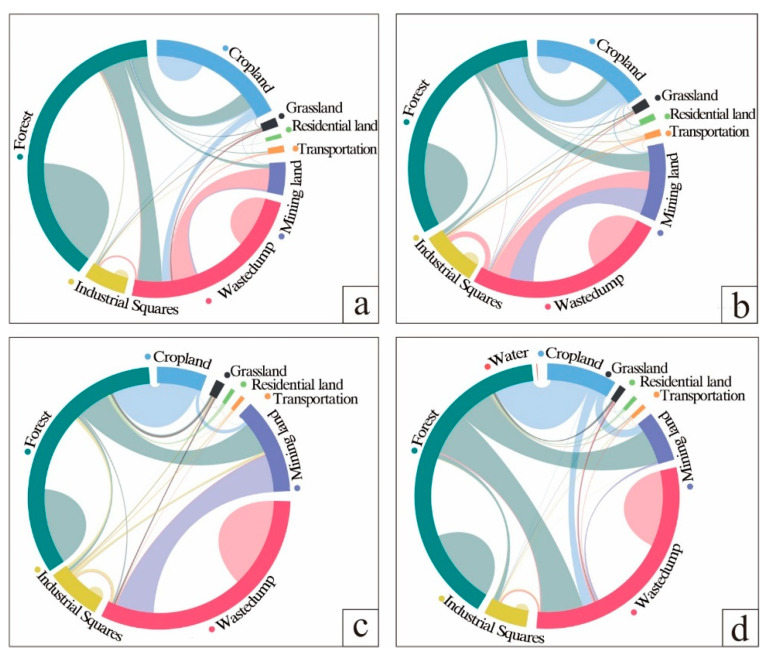
Relationship between land use type changes in Heidaigou mining. (**a**) 2006–2011; (**b**) 2011–2016; (**c**) 2016–2021; and (**d**) 2006–2021.

**Figure 9 ijerph-20-04394-f009:**
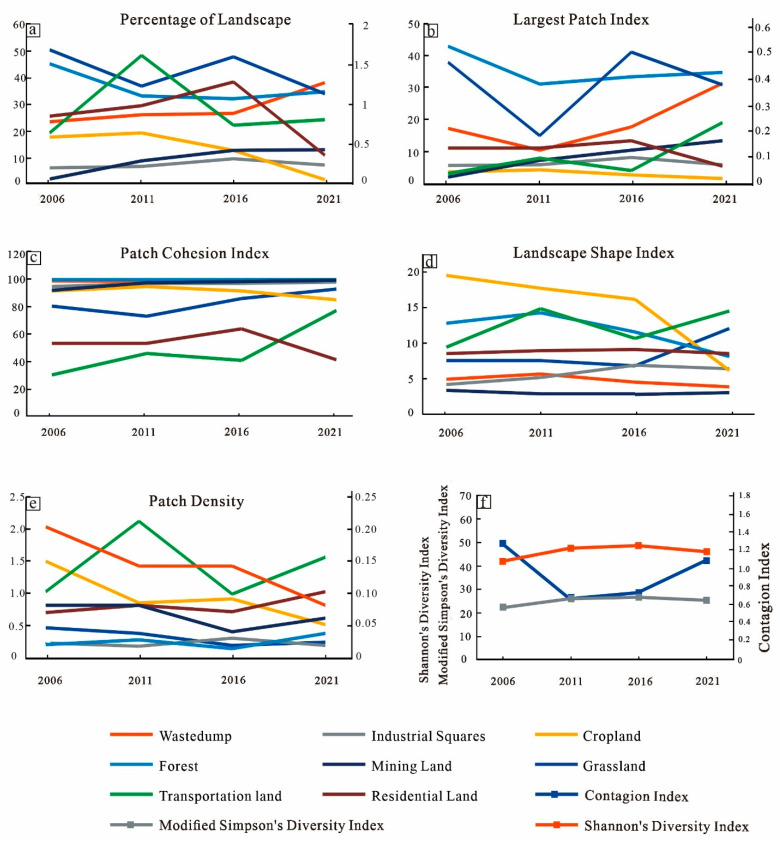
Characteristics of changes in landscape pattern index in the Heidaigou mining area. For class metrics: (**a**) The percentage of landscape (PLAND); (**b**) The largest patch index (LPI); (**c**) The patch cohesion index (COHESION); (**d**) The landscape shape index (LSI); (**e**) The patch density (PD). For landscape metrics: (**f**) The Shannon diversity index (SHDI), the contagion index (CONTAG), and the modified Simpson uniformity index (MSIEI).

**Figure 10 ijerph-20-04394-f010:**
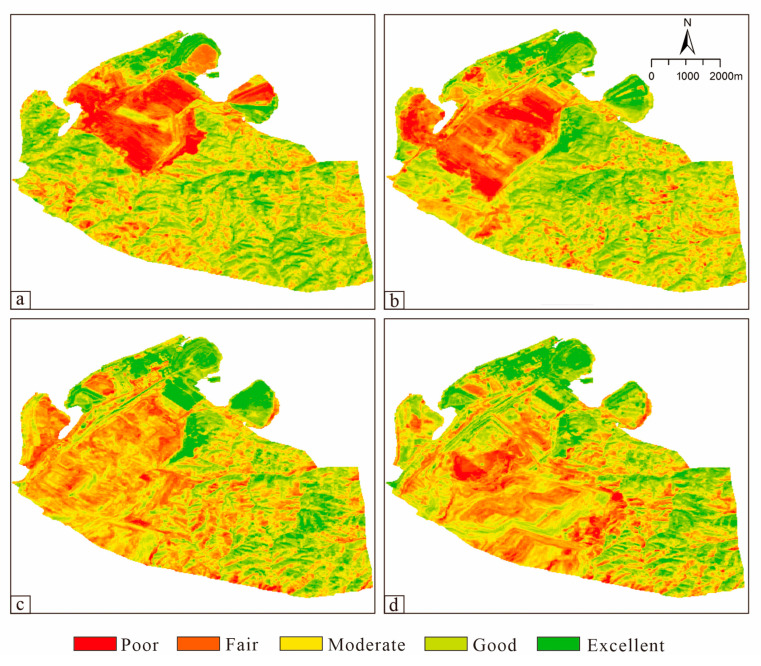
Distribution of RSEI grades at different periods in the mining area. (**a**) 2006, (**b**) 2011, (**c**) 2016, (**d**) 2021.

**Figure 11 ijerph-20-04394-f011:**
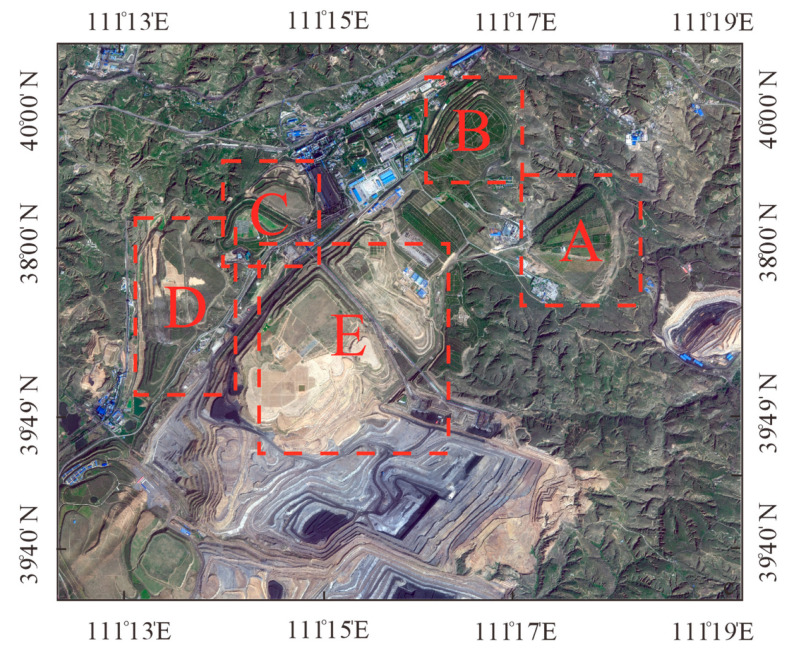
The location of each waste dump.

**Figure 12 ijerph-20-04394-f012:**
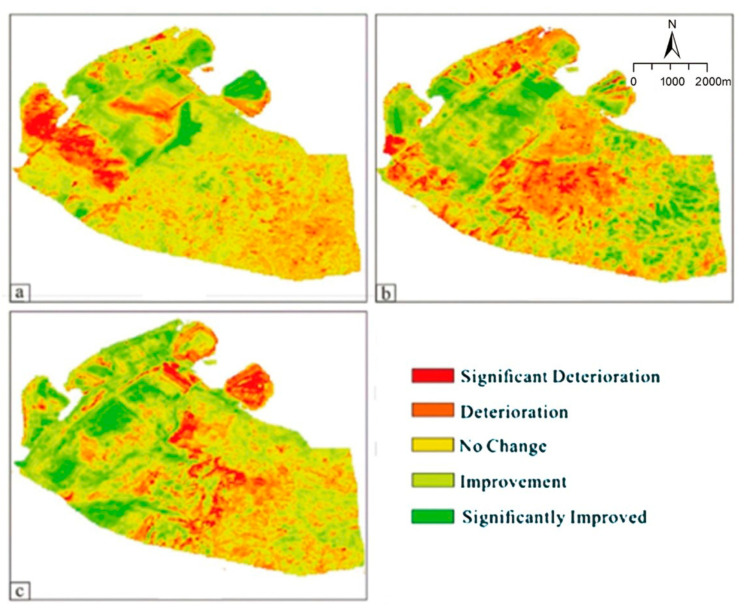
Characteristic map of eco–environmental quality evolution in different periods in mining area. (**a**) 2006–2011; (**b**) 2011–2016; (**c**) 2016–2021.

**Table 1 ijerph-20-04394-t001:** Kappa coefficients for LULC types from 2006 to 2021.

Year	Overall Classification Accuracy (%)	Kappa Coefficient
2006	88.39	0.87
2011	82.01	0.89
2016	83.03	0.80
2021	86.23	0.86
Average value	84.92	0.86

**Table 2 ijerph-20-04394-t002:** Heidaigou mining area 2006 to 2011 land use type transfer matrix (unit: hm^2^).

	2011	Grassland	Residential Land	Cropland	Transportation Land	Mining Land	Waste Dump	Industrial Squares	Forest	Total
2006										
Grassland		28.41	0.31	10.50	0.62	2.16	24.71	13.59	8.65	88.94
Residential Land		0.00	43.85	0.00	0.00	0.00	0.00	0.00	0.00	43.85
Cropland		0.00	1.54	894.70	0.93	6.79	157.51	19.15	36.75	1117.37
Transportation		0.00	0.00	0.00	27.18	0.00	1.85	1.54	2.16	32.74
Water		0.00	0.00	0.00	0.00	0.00	0.31	0.00	0.00	0.31
Mining Land		0.00	0.00	0.62	0.00	9.27	39.53	0.00	0.00	49.41
Waste Dump		19.46	0.00	0.00	24.71	419.71	983.95	32.74	19.77	1500.32
Industrial Squares		6.49	0.93	1.85	13.59	0.00	5.56	284.75	22.24	335.39
Forest		10.19	4.32	319.34	18.22	74.74	466.03	12.97	2069.20	2975.01
Total		64.55	50.96	1227.0	85.24	512.67	1679.45	364.73	2158.76	6143.35

**Table 3 ijerph-20-04394-t003:** Heidaigou mining area 2011 to 2016 land use type transfer matrix (unit: hm^2^).

	2016	Grassland	Residential Land	Cropland	Transportation Land	Mining Land	Waste Dump	Industrial Squares	Forest	Total
2011										
Grassland		23.78	0.00	2.78	0.62	0.00	5.25	18.22	13.90	64.55
Residential Land		0.00	36.75	2.16	0.00	0.00	0.00	4.01	8.03	50.96
Cropland		11.43	17.60	664.61	0.00	21.00	28.72	16.37	467.27	1227.00
Transportation land		4.94	0.31	5.56	8.65	11.12	14.52	24.40	15.75	85.24
Mining Land		0.62	0.00	0.00	0.00	45.09	466.03	0.93	0.00	512.67
Waste Dump		17.60	0.00	0.00	8.65	346.51	1193.65	104.39	8.65	1679.45
Industrial Squares		15.75	0.93	0.00	13.90	1.24	1.24	330.14	1.54	364.73
Forest		10.19	11.74	113.65	5.87	351.76	5.25	75.97	1584.32	2158.76
Total		84.31	67.33	788.76	37.68	776.72	1714.65	574.43	2099.46	6143.35

**Table 4 ijerph-20-04394-t004:** Heidaigou mining area 2016 to 2021 land use type transfer matrix (unit: hm^2^).

	2021	Grassland	Residential Land	Cropland	Transportation Land	Mining Land	Waste Dump	Industrial Squares	Forest	Total
2016										
Grassland		16.03	0.23	0.00	1.37	0.00	29.39	7.18	30.91	85.10
Residential Land		0.42	7.01	0.31	0.97	4.08	0.00	0.17	54.92	67.88
Cropland		23.50	5.08	30.62	2.39	63.05	0.00	1.67	673.72	800.03
Transportation land		1.49	0.00	0.00	16.07	0.00	6.07	7.70	5.77	37.10
Mining Land		0.00	0.00	0.00	1.01	77.10	705.02	5.91	0.04	789.08
Waste Dump		6.39	0.00	0.00	6.66	12.95	1685.79	11.90	6.40	1730.09
Industrial Squares		22.47	2.62	0.00	20.24	52.19	50.10	300.83	58.47	506.92
Forest		324.12	2.93	17.38	4.21	597.55	26.11	54.83	1100.02	2127.16
Total		394.41	17.86	48.32	52.93	806.93	2502.48	390.18	1930.24	6143.35

**Table 5 ijerph-20-04394-t005:** Heidaigou mining area 2006 to 2021 land use type transfer matrix (unit: hm^2^).

	2021	Grassland	Residential Land	Cropland	Transportation Land	Mining Land	Waste Dump	Industrial Squares	Forest	Total
2006										
Grassland		9.09	0.23	0.04	1.97	0.00	19.90	4.41	35.22	70.86
Residential Land		0.01	4.51	0.31	0.00	5.97	0.00	0.21	33.34	44.35
Cropland		0.42	5.72	33.63	1.25	140.56	200.72	6.26	721.89	1110.45
Transportation land		0.40	0.27	0.26	0.59	10.69	5.12	0.66	14.07	32.07
Water		0.00	0.00	0.00	0.00	0.00	0.19	0.00	0.00	0.19
Mining Land		0.00	0.00	0.00	0.00	1.29	48.73	0.00	0.28	50.30
Waste Dump		31.99	0.00	0.00	11.50	0.00	1419.16	27.79	22.69	1513.12
Industrial Squares		1.07	1.85	0.10	13.25	2.83	23.12	261.09	14.78	318.10
Forest		17.27	5.28	5.92	12.19	645.59	787.96	92.55	1437.16	3003.92
Total		60.24	17.86	40.26	40.75	806.93	2504.90	392.97	2279.43	6143.35

**Table 6 ijerph-20-04394-t006:** Principal component analysis of indicators from 2006 to 2021.

Indicators	2006	2011	2016	2021
	PC1	PC1	PC1	PC1
NDVI	0.513	0.572	0.509	0.554
Wet	0.492	0.511	0.521	0.516
NDSI	−0.541	−0.636	−0.599	−0.612
LST	−0.423	−0.245	−0.312	−0.383
Eigenvalue	0.051	0.059	0.063	0.059
Percent eigenvalue %	70.32	71.56	75.36	74.68

**Table 7 ijerph-20-04394-t007:** Area and percentage change of each RSEI level in different years.

RSEI Level	2006		2011		2016		2021	
	Area (hm^2^)	Pct. (%)	Area (hm^2^)	Pct. (%)	Area (hm^2^)	Pct. (%)	Area (hm^2^)	Pct. (%)
Poor	603.92	9.83%	402.49	6.55%	214.00	3.48%	226.29	3.68%
Fair	851.46	13.86%	846.29	13.78%	2322.42	37.80%	1993.08	32.44%
Moderate	2552.47	41.55%	2432.28	39.59%	2017.47	32.84%	1964.94	31.98%
Good	1827.55	29.75%	2219.82	36.13%	1099.58	17.90%	1204.96	19.61%
Excellent	307.95	5.01%	219.64	3.58%	463.99	7.55%	728.71	11.86%

**Table 8 ijerph-20-04394-t008:** The mean value of RSEI of the different years.

Year	2006	2011	2016	2021
RSEI	0.5002	0.5005	0.4993	0.5009
LST	0.4999	0.5006	0.5005	0.4998
NDVI	0.5002	0.4989	0.5003	0.5006
WET	0.5003	0.5005	0.5011	0.5006
NDBSI	0.4998	0.4996	0.5013	0.5000

## Data Availability

Not applicable.

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
