# Peer review of "The Evolution of Landscape Patterns and Its Ecological Effects of Open-Pit Mining: A Case Study in the Heidaigou Mining Area, China"

_ijerph, 2023, doi:10.3390/ijerph20054394_

Round 1
Reviewer 1 Report
The Heidaigou open-pit coal mine was taken as the research object to correctly understand and evaluate the impact of land use changes on landscape pattern and ecological environment, and the dynamic changes of land use and landscape pattern evolution characteristics of the mine area were analyzed based on medium and high-resolution remote sensing images in 2006, 2011, 2016 and 2021. The remote sensing ecological index was constructed to evaluate the ecological environment quality of the mining area in the past 15 years. The research result is of certain practical significance. But some figures do not conform to the specifications, the font size is too small to see clearly. Accept after minor revision.
Reviewer 2 Report
Taking your time to prepare it for a new submission is recommended. So far, there are a number of flaws in the manuscript, and the Guide for authors can be helpful in resolving these flaws before resubmitting the manuscript.
The following notes provide an overview of the manuscript's current status.
- Abstract contains 200 words max (https://www.mdpi.com/journal/ijerph/instructions#preparation)
- Line 33: too much decimals
- I suggest to insert at line 79: Minelli, A., Felice, E., Marinelli, E., Pasini, I., & Neri, D. (2021). Estimating leaf area indexes for 10 years of simultaneous or gradual renewal of an exhausted poplar tree avenue doi:10.17660/ActaHortic.2021.1331.40 Retrieved from www.scopus.com
- Reference 2 is missing from the text
- Wrong DOI for references [20] and [53]
- References [11], [36], [37], [38], [39], [40], [41] are incomplete
- A general language revision is required.
- It's advisable to implement and diversify the references
- Results should be extensively revised and synthesized to highlight the outcomes of this work.
Reviewer 3 Report
The introduction, methods and results sections are in good shape with relatively minor editorial changes. The author tend to hyperbolize statements and this needs to be changed to something more supportable (e.g., saying something is the most or best without support). However, the discussion and conclusions sections are completed inadequate and need to be rewritten. The discussion section mainly restates the results and does nothing to put the manuscript in context relating to its importance locally to the region or the importance of the work related to the scientific field in general. The conclusions section again restate results and draws no conclusions.
Minor editorial comments:
Line 16: Change "taken as the research object, and" to "used to assess"
Lines 25-26: Delete "At the classic level," and capitalize Patch
Lines 27-28: Delete ", the landscape pattern index type", change "has" to "have".
Lines 28-29: Delete "At the landscape level" and capitalize The
Line 48: Change "time" to "effort"
Line 57: Change "the most notable" to " a notable source of"
Line 58: Change "the most" to "a major source of"
Line 61-67: Don't use "it" repetitively
Lines 76-79: Place [22] after Garai et al. rather than at end of sentence
Lines 86-87: Place [31] after Xiong et al. rather than at end of sentence
Line 87: Define acronyms when first used (GEE)
Line 93: Change ". First, in" to ": (1)"
Line 95: Change ", Second, the" to "; (2)"
Line 96: Change ". Third, the" to "; and, (3)"
Figures 1a and 1b are pretty faint and difficult to see
Reviewer 4 Report
I attach the file.

Round 2
Reviewer 3 Report
Authors have addressed reviewers' comments